# The Role of miRNAs in Immune Cell Development, Immune Cell Activation, and Tumor Immunity: With a Focus on Macrophages and Natural Killer Cells

**DOI:** 10.3390/cells8101140

**Published:** 2019-09-24

**Authors:** Shi Jun Xu, Hong Tao Hu, Hai Liang Li, Suhwan Chang

**Affiliations:** 1Department of Radiology, The Affiliated Cancer Hospital of Zhengzhou University, Henan Cancer Hospital, Zhengzhou 450008, China; lsxusj@126.com; 2Department of Minimal Invasive Intervention, The Affiliated Cancer Hospital of Zhengzhou University, Henan Cancer Hospital, Zhengzhou 450008, China; hht19761213@163.com; 3Department of Biomedical Sciences, University of Ulsan College of Medicine, Seoul 05505, Korea

**Keywords:** miRNA, tumor immunity, macrophages, natural killer cells

## Abstract

The tumor microenvironment (TME) is the primary arena where tumor cells and the host immune system interact. Bidirectional communication between tumor cells and the associated stromal cell types within the TME influences disease initiation and progression, as well as tumor immunity. Macrophages and natural killer (NK) cells are crucial components of the stromal compartment and display either pro- or anti-tumor properties, depending on the expression of key regulators. MicroRNAs (miRNAs) are emerging as such regulators. They affect several immune cell functions closely related to tumor evasion of the immune system. This review discusses the role of miRNAs in the differentiation, maturation, and activation of immune cells as well as tumor immunity, focusing particularly on macrophages and NK cells.

## 1. Introduction

Bidirectional communication between cells and their microenvironment is crucial for normal tissue homeostasis as well as tumor growth [1]. The interactions between tumor cells and associated stromal cells represent an especially dynamic relationship that affects disease initiation and progression [2]. The tumor microenvironment (TME) is the primary site where tumor cells and the host immune cells interact. There are multiple stromal cell types within the TME. Among them, macrophages and natural killer (NK) cells are the most prevalent. They not only serve as the first barrier against pathogen infection, but also play vital roles in tumor progression.

Macrophages are cells involved in innate immunity, and they are a key player in linking innate and adaptive immunity. During pathogen infection or in the presence of a tumor cell-specific antigen, pattern-recognition receptors (PRRs) expressed in macrophages recognize components of the pathogen or the antigen and produce type I interferon (IFN I). This induces IFN-stimulated genes (ISGs), causes the expression of immune response mediators (cytokines and chemokines), and enhances antigen presentation against pathogen infection, tumor growth, or tumor invasion [3]. NK cells are also effectors of the innate immune system. They provide an early cellular defense against pathogens or tumor cells by synthesizing cytokines and chemokines, and causing cytotoxicity in tumor or infected cells to limit their spread and subsequent tissue damage. In addition, recent studies highlight that NK cells also regulate several types of immune cells, including macrophages; thus, NK cells also play a role in controlling inflammatory and autoimmune disorders [4]. However, once tumor cells or infected cells circumvent the first barrier of immune defense, a tumorigenic primary niche will form and develop further. Meanwhile, the TME converges to re-educate stromal cells to support pathogen infection and tumor progression (macrophages are the most prevalent) [1], which is partially mediated by microRNAs (miRNAs) and their downstream transcriptional factors [5].

MicroRNAs (miRNAs) are a large family of small (~23 nt) endogenous non-coding RNAs. They negatively regulate gene expression at the post-transcriptional level by binding to the 3′-untranslated region (UTR) of target mRNAs, which degrades the mRNAs or represses their translation [6]. Decades of studies have demonstrated that miRNAs participate in nearly every biological process, including immune cell development and maturation, as well as tumor immunity.

As differences in the TME among different cancer types are more fully understood [7], studies examining the relationship between the TME and stromal cells have increased in recent years. Herein, we summarize the functions of miRNAs in the development, maturation, activation, and tumor immunity with a specific focus on macrophages and NK cells.

## 2. The Role of miRNAs in Macrophages

Macrophages are essentially present in all tissues, and they are crucial effectors of wound healing, homeostasis, cancer, and immune responses [1,8]. Most macrophages derive from hematopoietic stem cells (HSCs). They sequentially differentiate into lymphoid-myeloid progenitors (LMP), granulocyte-macrophage progenitors (GMP), and then monocytes, which migrate to various tissues to become mature macrophages [1,9,10]. However, tissue-resident macrophages are derived from yolk-sac-derived erythro-myeloid progenitors [11,12]. Diversity and plasticity are the foremost characteristics of cells derived from a monocyte–macrophage lineage, especially for tumor-associated macrophages (TAMs) [13]. TAMs can rapidly alter their polarization states to accommodate different tissue microenvironment, which explains why macrophages have such disparate roles during normal homeostasis and tumorigenesis [1]. Classically activated (M1) macrophages display anti-tumorigenic activities by producing type I pro-inflammatory cytokines and participating in antigen presentation [14]. In contrast, alternatively activated (M2) macrophages secrete type II cytokines, which improve anti-inflammatory responses and display pro-tumorigenic activities [14]. M2 macrophages are further divided into four subtypes: M2a, M2b, M2c [15], and M2d (TAMs belong to this subtype) [16]. Decades of research show that several transcriptional factors control the activation of these various phenotypes, and the activities of these transcriptional factors are partially controlled by miRNAs.

A specific set of miRNAs are implicated in hematopoietic stem cell (HSC) differentiation and maturation. Conditional ablation of Dicer in HSCs showed that mmu-miR-29a, mmu-miR-126, *mmu-miR-130a*, *mmu-miR-155*, and *mmu-miR-125a/b* control the differentiation of HSCs by targeting different genes [17]. For example, *mmu-miR-125a* significantly increased stem cell quantities by targeting *BAK1* [18], and *mmu-miR-126* targeted *PI3K/AKT/GSK3β* to exert the same function [19]. Transcription factor PU.1 is a key factor in lympho-myeloid development and stimulates the differentiation of HSCs into LMPs by inducing the expression of four miRNAs (*mmu-miR-146a*, *mmu-miR-342*, *mmu-miR-338*, and *mmu-miR-155*). By transiently occupying the binding sites within regulatory chromatin regions adjacent to their genomic coding loci, PU.1 induces HSCs to differentiate into macrophages. In addition, in vivo development assays conducted in mouse and zebrafish models show that *miR-146a* directs the selective differentiation of HSCs into functional macrophages [20]. Meanwhile, PU.1 suppresses the expression of the *mmu-miR-17-92 cluster* to force the premature differentiation of stem and progenitor cells into a myeloid lineage. They likely block the c-MYC-mediated proliferation of progenitor cells to ensure correct terminal differentiation [21]. The CCAAT/enhancer-binding protein alpha (C/EBPα) plays an essential role in differentiating LMPs into GMPs by directly binding to the promoter of *hsa-miR-223* to increase its expression, which promotes granulocytic differentiation [22]. On the other hand, *hsa-miR-424* promotes the differentiation of GMPs into monocytes in humans [23], while *miR-21* and *miR-196b* play the same roles in both human and mouse models [24]. Recent research has shown that the knockdown of *hsa-miR-128a* induces Lin28a expression and reverts myeloid differentiation blockage in acute myeloid leukemia [25], but *hsa-miR-181a* reduces granulocytic and macrophage-like differentiation as well as hematopoietic stem/progenitor cell accumulation by targeting and down-regulating the expression of *PRKCD*, *CTDSPL*, and *CAMKK1* [26]. Moreover, *hsa-miRs-17-5p/20a/106a clusters* suppress blast proliferation and inhibit monocyte differentiation and maturation by targeting *AML1* [27]. Furthermore, next-generation SOLiD sequencing shows that *hsa-miR-106-3p*, *hsa-miR-132-3p*, *hsa-miR-335-5p*, *hsa-miR-34a-5p*, *hsa-miR-362-3p*, and *hsa-miR-424-5p* are up-regulated in macrophages when compared to monocytes [28], which implies that these miRNAs are involved in the maturation of macrophages.

miRNAs are also involved in macrophage polarization and activation. Recently, it was discovered that many genes and their related signaling pathways function in the transition of macrophage phenotypes. These transcription factors include cytokines, kinases, phosphatases, receptors, and miRNAs [13,29,30]. To investigate the role of miRNAs in macrophage phenotype switching, Lu et al. investigated the time-dependent miRNA–mRNA transcriptomic changes between the M1 and M2 transitions [31]. They found that *mmu-miR-155-3p*, *mmu-miR-155-5p*, *mmu-miR-145-3p*, and *mmu-miR-9-5p* are the four highest expressed miRNAs in M1 macrophages, and that *mmu-miR-27a-5p*, *mmu-let-7c-1-3p*, *mmu-miR-23a-5p*, and *mmu-miR-23b-5p* are the four highest expressed miRNAs in M2 macrophages derived from the bone marrow of mice. In addition, they found that *mmu-miR-1931*, *mmu-miR-3473e*, and *mmu-miR-5128* function as early-response miRNAs. However, the role of miRNAs in human macrophage polarization at different times is still unclear. Other miRNAs involved in macrophage polarization and activation are shown in Table 1 and Figure 1.

Tumor-derived miRNAs play crucial roles in macrophage functions and tumor immunity. For example, *mmu-miR-142-3p* is down-regulated in tumor filtered myeloid CD11b^+^ cells, promotes macrophage differentiation, and determines the acquisition of their immunosuppressive function in tumors [32]. In a mouse breast cancer model, mmu-miR-155 is up-regulated in CD11c^+^ pro-inflammatory TAMs and actively mediates tumor immunity, especially during the early stages of breast carcinogenesis [33].

Virus-encoded or virus infection-induced miRNAs also regulate macrophage activities in the tumor microenvironment. BamHI fragment A rightward transcript (BART) miRNA derived from Epstein Barr Virus (EBV)-infected Akata-lymphoblastoid cell lines converts macrophages into TAMs by partially regulating TNF-α, IL-10, and arginase 1 (ARG1) expression [34]. Virus-encoded miRNAs (e.g., *miR-H1*, *miR-K12-3-3p*, *miR-UL-70-3p*, and *EBV-miR-BART11*) that are incorporated into macrophages alter cellular gene expression (including miRNA expression) and convert M1 stage macrophages into M2 stage macrophages, which facilitates tumor development and metastasis [34,35,36].

Finally, many miRNAs also suppress tumor immunity by blocking the expression of key regulators involved in the activation of innate immunity pathways. For example, Xu et al. showed that rhabdovirus infection significantly induced *miR-3750* expression in macrophages by targeting MAVS, which is an adaptor gene involved in RIG-I pathway activation [37]. However, some viral-encoded miRNAs contribute to tumor immunity. The H5N1 influenza virus-encoded miRNA *miR-HA-3p* promotes cytokine production in human macrophages by targeting poly(rC) binding protein 2 (PCBP2), which is a negative regulator of RIG-I-mediated antiviral innate immunity [38]. miRNAs involved in tumor immunity or immunity activation are summarized in Table 1.

**Table 1 cells-08-01140-t001:** A list of miRNAs involved in macrophage development, macrophage polarization, and tumor immunity.

Development and Maturation	Promotes M1	Suppresses M1	Promotes M2	Suppresses M2	Related to Tumor Immunity
*mmu-miR-29a* [17](+)	*hsa-miR-155* [39,40] *mmu-miR-155-3p/5p* [31]	*mmu-miR-124* [41]	*hsa-miR-27a* [40] *mmu-miR-27a-5p* [31]	*hsa-miR-130a* [42] *mmu-miR-130b* [43]	*hsa-miR-3570* [37](–)
*mmu-miR-126* [19](+)	*hsa-miR-125a/b-5p* [40] *mmu-miR-125a-3p* [31,44] *mmu- miR-125b* [45]	*hsa-miR-181a* [46] *mmu-miR-181a* [46]	*mmu-miR-23a/b-5p* [31]	*hsa-miR-27b* [47] *mmu-miR-27a* [48]	*hsa-miR-3614-5p* [49](–)
*mmu-miR-130a* [17](+)	*hsa-miR-29b* [40] *mmu-miR-29b-1-5p* [31]	*hsa-miR-9* [50]	*mmu-miR-188* [31]	*mmu-miR-21* [51]	*hsa-miR-29* [52](+)
*mmu-miR-155* [17,20](+)	*hsa-miR-145-5p* [28] *mmu-miR-145-5p* [31]	*mmu-let-7c* [53] *mmu-let-7d-5p* [54]	*mmu-let-7c-1-3p* [31] *mmu-let-7c* [53] *mmu-let-7d-5p* [54]	*mmu-miR-23a* [55]	*mmu-let-7d-5p* [54](–)
*mmu-miR-125a/b* [17,18](+)	*mmu-miR-147-5p/3p* [31]	*mmu-miR-210* [56]	*hsa-miR-26a* [40]	*hsa-miR-155* [57,58]	*mmu-miR-155* [33](+)
*mmu-miR-146a* [20](+) *hsa/mmu-miR-146a* [59](+)	*mmu-miR-9-5p/3p* [31]	*mmu-miR-93* [60]	*hsa-miR-146a/b* [40] *mmu-miR-146b* [61]	*hsa-miR-720* [62]	*mmu-miR-223* [63](+)
*mmu-miR-342* [20](+)	*mmu-miR-21* [51]	*hsa-miR-146b* [64] *mmu-miR-146b* [61]	*hsa-miR-222-3p* [40]	*mmu-miR-125a-3p* [31,44] *mmu- miR-125b* [45]	*hsa-miR-23a-3p* [65](–)
*mmu-miR-338* [20](+)	*mmu-miR-33* [66]	*mmu-miR-15a/16* [67]	*mmu-miR-127* [68]	*mmu-miR-26a* [69] *hsa-miR-26a/b* [70,71]	*hsa-miR-146a* [59](+)
*mmu-miR-17-92 cluster* [21] (–) *hsa-miR-17-5p-20a-106a-92* [27](–)	*mmu-miR-330-5p* [72]	*hsa-miR-30d-5p* [73]	*hsa-miR-181a* [46] *mmu-miR-181a* [46]	*hsa-miR-19a-3p* [74] *mmu-miR-19a-3p* [74]	*hsa-miR-17/20a/106a* [75](+)
*hsa-miR-223* [22](+)	*mmu-let-7e-3p* [31]	*hsa-miR-24* [76]	*hsa-miR-145-3p* [77]	*mmu-miR-33* [66]	*mmu-miR-142-3p* [32](+)
*hsa-miR-424* [23](+)	*mmu-miR-1931* [31]	*mmu-miR-223* [78]	*mmu-miR-223* [78]	*mmu-miR-330-5p* [72]	*hsa-miR-34a* [79](–)
*hsa/mmu-miR-21* [24](+)	*mmu-miR-3473e* [31]	*mmu-miR-21* [80,81]	*hsa-miR-181b* [82] *mmu-miR-181b* [82]	*mmu-miR-127* [68]	*hsa-miR-195-5p* [83](+)
*hsa/mmu-miR-196b* [24](+)	*mmu-miR-5128* [31]		*hsa-miR-103a* [84]	*hsa-miR-935* [85]	*hsa-miR-301a* [86](–)
*hsa-miR-128a* [25](–)	*mmu-miR-222-5p* [31]		*hsa-miR-30d-5p* [73]	*mmu-miR-148a-3p* [87]	*hsa-miR-375* [88](+)
	*mmu-miR-3473b* [31]		*mmu-miR-124* [89]	*mmu-miR-511-3p* [90]	*miR-HA-3p* [38](+)
*mmu-miR-142-3p* [32](–)	*hsa-miR-199a-5p* [91]		*mmu-miR-142-3p* [32]	*mmu-miR-378-3p* [92]	*BART miRNA* [34](–)
	*mmu-miR-127* [68]		*hsa-miR-940* [93]	*hsa-miR-98* [94]	*ebv-miR-BART11* [36](–)
*hsa-miR-106-3p* [28](+)	*mmu-miR-148a-3p* [87]		*hsa-miR-24* [76]	*hsa-miR-195-5p* [83]	
*hsa-miR-132-3p* [28](+)	*hsa-miR-130a* [42] *mmu-miR-130b* [43]		*hsa-miR-202-5p* [95]	*hsa-miR-199a-5p* [91]	
*hsa-miR-335-5p* [28](+)	*hsa-miR-27b* [47] *mmu-miR-27a* [48]		*hsa-let-7b* [96]		
*hsa-miR-34a-5p* [28](+)	*mmu-miR-26a* [69]		*hsa-miR-34a* [79]		
*hsa-miR-362-3p* [28](+)	*miR-HA-3p* [38]		*hsa-miR-301a* [86]		
*hsa-miR-424-5p* [28](+)			*mmu-miR-21* [80,81]		
*hsa-miR-223/15a/16* [97](–)			*BART miRNAs* [34]		
			*miR-H1* [34]		
			*miR-K12-3-3p* [34]		
			*miR-UL-70-3p* [34]		
			*ebv-miR-BART11* [36]		

Note: (+), promote the process; (–), suppress the process.

## 3. The Role of miRNAs in NK Cells

Natural killer (NK) cells are cytotoxic innate lymphoid cells and are critical mediators of early host defense against pathogen infection, immune homeostasis, and tumor surveillance [98]. NK cells originate in bone marrow and complete their maturation in peripheral organs, which leads to their phenotypical and functional heterogeneity [99]. The amount and type of receptors on the surface of NK cells determine their functionality [100,101]. Based on the number of CD56 and CD16 surface markers, human NK cells are divided into two subsets: CD56^bright^/CD16^–/dim^ and CD56^dim^/CD16^bright^. The latter is the main form of circulating NK cells [102]. CD56^bright^ cells regulate the activation and function of NK cells, as well as other immune cells, by secreting cytokines such as IFN-γ and TNF-α. However, CD56^dim^ cells release lytic molecules such as perforin and granzyme B, to exert highly cytotoxic effects. CD56^dim^ cells are also crucial for antibody-mediated cytotoxicity [99].

The development and maturation of human NK cells can be divided into two primary stages. In stage 1, NK cells are derived from bone marrow HSCs and progress through common lymphoid progenitors (CLPs), CD34 pro-NK, CD122 pre-NK, and committed immature NK cells (iNKs) stages, which finishes their NK lineage commitment. At this time, NK cells lose the capacity for T-cell or dendritic cells (DC) development [103]. In stage 2, iNK cells move to peripheral tissues such as the spleen or liver to complete their differentiation and maturation. In those peripheral tissues, iNKs convert into functional CD56^bright^ or CD56^dim^ NK cells [104]. In contrast, the development of mouse NK cells is different from human NK cells in several ways. CLPs firstly differentiate into common innate lymphoid progenitors (CILPs) with the help of transcriptional factor Nifil3. They then gradually develop into NK-cell precursors (NKPs), iNKs, and mouse mature NK cells (mNKs) [105]. Many transcriptional factors are necessary during the development and maturation of NK cells [105]. Accordingly, several studies have revealed that miRNAs are important in the regulation of fundamental NK cell processes such as activation, cytotoxicity, proliferation, development, and maturation by targeting the receptors or factors involved in transcriptional expression [106,107,108]. Through next-generation sequencing (NGS), many miRNAs have been discovered, and their roles in regulating NK cell development and maturation as well as disease progression have been verified.

A recent microarray study compared the expression of miRNA between mouse splenic NK cells (NK1.1^+^TCRβ^−^) and human peripheral blood NK cells (CD56^+^CD3^–^). Aimee et al. discovered the 14 conserved miRNAs with the highest expression levels in both groups: *miR-150*, *miR-23b*, *miR-29a*, *miR-23a, miR-16*, *miR-21*, *let-7a*, *let-7f*, *miR-24*, *miR-15b*, *miR-720*, *let-7g*, *miR-103*, and *miR-26a* [109]. This implies that these miRNAs should be crucial for NK cell functionality. Other previous studies have revealed that *mmu-miR-150* and *hsa-miR-181* promote the differentiation of pre-NKs into iNKs by targeting c-Myb or the Notch signaling inhibitor Nemo-like kinase (NLK), respectively [110,111]. Furthermore, *mmu-miR-150* and *mmu-miR-15/16* assist in the maturation of iNKs into mNKs by targeting the same gene, c-Myb [111,112]. During the activation of mature NKs, hsa-miR-155, *mmu-miR-150*, *mmu-miRs-15/16*, *hsa-miR-181*, and *mmu-miR-29* suppress IFN-γ production in CD56^bright^ NK cells. Mechanistically, *mmu-miRs-15/16* and *mmu-miR-29* directly target IFN-γ 3′UTR [113,114], and *hsa-miR-181* indirectly represses upstream targets [110] to reduce IFN-γ translation. However, *miR-155* regulates IFN-γ production in human and mouse NK cells by modulating the expression of the phosphatase SHIP-1, inhibiting T-bet/Tim-3, or by decreasing the activation of several signaling pathways such as those involving PI3K, NF-κB, and calcineurin [115,116,117]. *hsa-miR-181* promotes IFN-γ production in primary CD56(+) NK cells [110]. *miR-150* not only targets c-Myb to facilitate NK cell development and maturation [111], but it also represses PIK3AP1 and AKT2, which is a part of the PI3K-AKT pathway, and up-regulates Bim and p53 to assist in NK cell apoptosis [118]. In addition, *miR-223*, *miR-27a-5p*, *miR-150*, *miR-378*, and *miR-30e* suppress the cytotoxic capabilities of CD56^dim^ NK cells. Mechanistically, mmu-miR-233 directly binds to the 3′UTR of granzyme B [119], and *hsa-miR-150* targets the perforin 3′UTR [120]. hsa-miR-27a*-5p [121], hsa-miR-378, and *hsa-miR-30e* [122] repress the 3′UTR of granzyme B and perforin to repress the cytotoxic capacity of NK cells. miRNAs functioning in NK cell development, maturation, and activation are listed in Table 2 and Figure 2.

Pathogen- and tumor-induced miRNAs also regulate NK cell activities in the tumor microenvironment. For example, Cheng et al. found that Hepatitis C virus (HCV) infection down-regulates *hsa-miR-155* in NK cells. The down-regulation of *hsa-miR-155* releases T-bet/Tim-3, which suppresses IFN-γ production and leads to HCV evading immune clearance [117]. Importantly, TGF-β, a key mediator in the TME, post-transcriptionally increases mature *hsa-miRNA-1245* expression. This miRNA suppresses *NKG2D* expression, which blocks NKG2D-mediated immune responses in NK cells and supports the TME [123]. TGF-β also induces *hsa-miR-183* to abrogate the tumor cell-killing function of NK cells by targeting DNAX activating protein 12kDa (DAP12) [124]. In addition, *hsa-miR-146a* intervenes in NK cell IFN-γ synthesis by down-regulating RAK1 and TRAF6 expression [125]. *hsa-miR-519a-3p* impairs NK cell function by down-regulating the NKG2D ligands ULBP2 and MICA on the surface of tumor cells, and also affects granzyme B-induced apoptosis and caspase-7 activation in breast cancer [126]. More information about these miRNAs is summarized in Table 2 and Figure 2.

**Table 2 cells-08-01140-t002:** A list of miRNAs involved in NK cell development, NK cell activation, and tumor immunity.

Development and Maturation	Classical Activation		NK Cell-Related Tumor Immunity Escape
	IFN-γ Production	Cytotoxicity	
*mmu-miR-155* [127](+)	*hsa-miR-155* [115,116,117](+)	*hsa-miR-1245* [123](–)	*hsa-miR-155* [117](–)
*mmu-miR-150* [111](+)	*hsa-miR-146a* [125](-)	*hsa-miR-183* [124](–)	*hsa-miR-1245* [123](+)
*hsa-miR-181* [110](+)	*hsa-miR-122/15b* [128](+)	*hsa-miR-519a-3p* [126](–)	*hsa-miR-183* [124](+)
*mmu-miR-15/16* [112](+)	*mmu-miR-155* [127](+)	*mmu-miR-223* [119](–)	*hsa-miR-519a-3p* [126](+)
*hsa-miR-29b* [129](–)	*mmu-miR-15/16* [113](-)	*hsa-miR-150* [120](–)	*hsa/mmu-miR-146b-5p* [130](–)
*hsa-miR-218* [131](–)	*mmu-miR-150* [111](–)	*hsa-miR-27a*-5p* [121](–)	*hsa-miR-296-3p* [132](+)
	*hsa-miR-181* [110](–)	*hsa-miR-378* [122](–)	*hsa-miR-146a* [133](+)
	*mmu-miR-29* [114](–) *hsa-miR-29a* [134](–)	*hsa-miR-30e* [122](–)	*hsa-miR-376a(e)* [135](+)
	*hsa-miR-362-5p* [136](+)	*hsa-miR-20a* [137](–)	*hsa-miR-186* [138](–)
	*hsa-miR-302c/520c* [139](–)	*hsa-miR-362-5p* [136](+)	*hsa-miR-122-5p* [138](–)
	*hsa-miR-122-5p* [140](+)	*hsa-miR-30c-1-3p* [141](+)	*hsa-miR-222-3p* [138](–)
	*hsa-miR-132* [142](–) *hsa-miR-212* [142](–) *hsa-miR-200a* [142](–)	*hsa-miR-146a* [133](–)	*hsa-miR-29b* [129](+) *hsa-miR-29* [143](–) *mmu-miR-29b* [144](+)
		*hsa-miR-302c/520c* [139](–)	*hsa-miR-519a-3p* [126](+)
		*hsa-miR-186* [138](+)	*hsa-miR-141* [145](+)
		*hsa-miR-519a-3p* [126](–)	*hsa-miR-548q* [146](–)
		*hsa-miR-23a* [147](–)	*hsa-miR-23a* [147](+)
		*hsa-miR-10b* [148](–)	*hsa-miR-17-92* [149](+) *hsa-miR17/20a* [150](–)
		*hsa-miR-506* [151](+)	*hsa-miR-373* [152](+)
		*hsa-miR-548q* [146](+)	*hsa-miR-23b* [148](+)
		*hsa-miR-152* [153](+)	*hsa-miR-27a-5p* [154](+)
		*mmu-miR-18a* [155](–)	*hsa-miR-561-5p* [156](+)
		*hsa-miR-132/212/200a* [142](–)	*hsa-miR-132/212/200a* [142](+)
		*ebv-miR-BART7* [157](–)	*hsa-miR-34a/c* [158](–)
		*miR-M23-2* [159](–) *miR-m21-1* [159](–)	*hsa-miR-30e* [160](+)
		*miR-UL112* [161](–)	*miR-J1-3p* [162](–)
		*miR-J1-3p* [162](+)	*hcmv-miR-UL112* [163](+)
		*hcmv-miR-UL112* [163](–)	*miR-K12-7* [164](+) *miR-BART2-5p* [164](+)
			*EBV-miR-BART20-5p* [165](+) *EBV-miR-BART8* [165](+)
			*HSV1-miR-H8* [166](+)
			*ebv-miR-BART7* [157](+)
			*kshv-miR-K12-1* [167](+)

Note: (+), promote the process; (–), suppress the process.

## 4. Conclusions and Further Perspectives

Extensive studies have contributed to the characterization of the TME and improved our understanding of cancer. Only now are we beginning to understand how the stromal cell-mediated immune response determines cancer initiation and progression. There are many factors involved in this process, including miRNAs. miRNAs not only regulate the development and maturation of immune cells, but they also control the activation of immune cells and their subsequent actions as pro- or anti-tumor factors (Figure 1 and Figure 2).

All known major miRNAs involved in macrophage development, polarization, and tumor immunity are summarized in Table 1 and Figure 1. By analyzing these data, we find that miRNAs such as *miR-146a/b* [59,61,64], *miR-17-92 cluster* [21,27], *miR-181a/b* [46,82], and *miR-155* [31,40], whether they are derived from mouse or human models, usually have similar effects on macrophage development and polarization. Many miRNAs can regulate M1 and M2 states at the same time; for instance, *miR-181a* [46], *miR-146b* [61], and *mmu-let-7c/d* [53,54] simultaneously promote M2 activation and suppress M1 activation. *miR-27a/b* and *miR-125a* promote M1 activation while inhibiting M2 activation [31,40]. It is worth noting that the same miRNA may have conflicting functions in controlling macrophage transitions. For example, one report mentions that *mmu-miR-21* increases M1 activation and inhibits M2 functioning [51]. However, according to other results, *mmu-miR-21* restrains M1 activation and promotes M2 macrophage polarization [80,81]. These discrepancies might exist because *miR-21* may exert different influences on macrophage polarization depending on the cell type, cancer type, and TME, since host tissues contain various types and quantities of stromal cells, which are determinants of tumor immunity [7]. Furthermore, the diversity and complexity of the TME may be the main reason why so many miRNAs are necessary and why a single, key miRNA that regulates macrophage functioning has not been found. Therefore, to further clarify the exact role of each miRNA in tumor immunity, more studies examining different cancers, organs, and tissues are necessary to estimate the multiple types of stromal cells functioning in each system.

Of note, Table 2 only lists a handful of miRNAs that contribute to NK cell development. This may be due, in part, to discrepancies in the microarray and small RNA-seq data that were used to uncover miRNAs involved in NK cell development or activation [108,109,119,122,168]. For example, miR-150 displays the greatest expression in the human microarray data [109], but it is not even among the top 15 [122] or top 25 miRNAs [108] being expressed, according to the NGS data. This might be due to the variability between the two methods, or the small RNA library construction approaches. Furthermore, the *miR-150* profile is distinct between human and mouse NK cells when using NGS, which may be due to interspecies or experimental replications variation. Although the same method was used to obtain human NK cells (CD56^+^CD3^–^), *miR-150* expression results were inconsistent between NGS and microarray studies, which is likely caused by variability between the two methods. Fehniger’s group proved that all miRNAs detected by SOLiD can be verified using qRT-PCR or a microarray [119]. However, approximately 25% of miRNAs detected by NGS cannot be verified by qRT-PCR or a microarray [119], implying that NGS may be uncovering many potential novel miRNA precursor genes. Therefore, more reasonable and more reliable detection methods are urgently needed to precisely interpret the role of miRNA in NK cell biology.

In this review, we described the roles of miRNAs in the development, maturation, activation, and tumor immunity of macrophages and NK cells (Table 1 and Table 2). We also mentioned the different activities displayed by human miRNAs and mouse miRNAs in all of the above processes. Our data show that certain miRNAs have different roles in various cell functions. For example, both human and mouse *miR-146a* inhibit IFN-γ production and the cytotoxicity of NK cells, and associate with tumor cells to escape immune surveillance. However, *miR-146a* promotes the development and maturation of human macrophages, and it promotes immune system activities. This phenomenon implies that the effectiveness of one miRNA treatment is limited, and this may be the reason why only a handful of miRNAs are successful at treating cancer, even though a large number of miRNAs participate in immune cell functions. Some miRNAs such as *hsa-miR-181a*, *mmu-miR-150*, and *mmu-miR-155* influence multiple stages in various immune cells, making them good candidates for drug development. Considering that several miRNAs are potential therapeutic options for the treatment of different cancers, it is likely that a cocktail of miRNAs, instead of a single miRNA, may be more effective as a therapeutic option. Importantly, more studies are needed to discover miRNAs that may be involved in stromal cell development and maturation, and enhance the effect of immunotherapy by attracting more functional immune cells. Finally, more effective approaches such as cross-linking immunoprecipitation (CLIP) need to be developed or applied to accurately reveal miRNAs that influence stromal cells and tumor immunity in the tumor microenvironment. 

## Figures and Tables

**Figure 1 cells-08-01140-f001:**
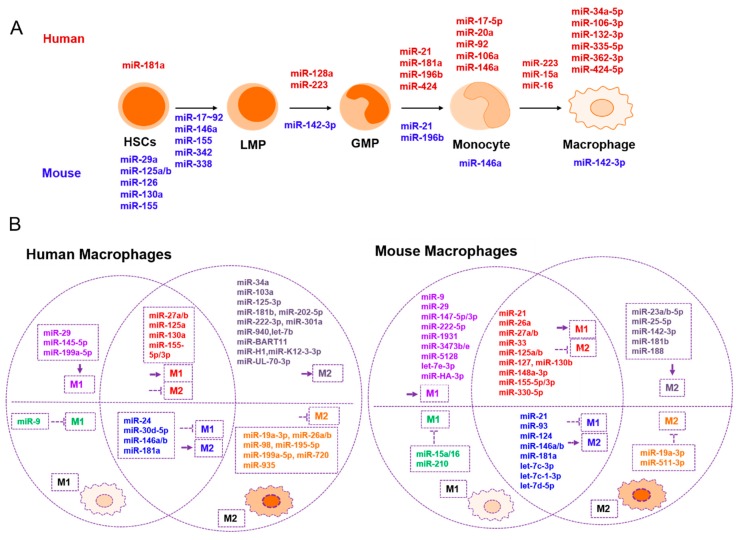
miRNAs are involved in macrophage development, polarization, and tumor immunity. (**A**) miRNAs involved in mouse and human macrophage development and maturation. miRNAs listed without arrows participate in each step of cell differentiation or maturation, while miRNAs listed with arrows function in the developmental transition. (**B**) The role of miRNAs in classical M1 macrophage activation or M2 macrophage alternative activation in humans and mice. Different colors indicate the different roles that miRNAs play in macrophage polarization. HSCs, hematopoietic stem cells; LMP, common lymphoid progenitor; GMP, granulocyte-macrophage progenitor; M1, classically activated macrophages; M2, alternatively activated macrophages.

**Figure 2 cells-08-01140-f002:**
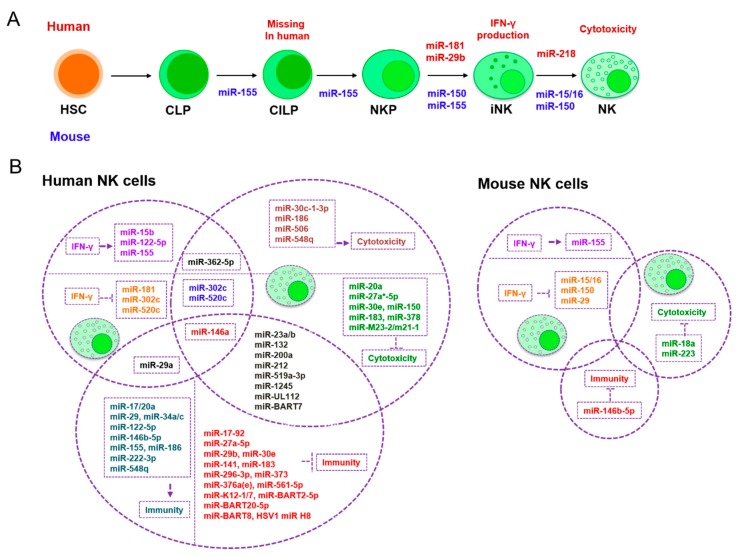
miRNAs involved in NK cell development, NK cell activation, and tumor immunity. (**A**) miRNAs involved in mouse and human NK cell development and maturation. miRNAs listed with arrows regulate each developmental transition of NK cells. Note that human NK cell development, unlike mouse NK cell development, lacks a CILP stage. (**B**) miRNAs involved in IFN-γ production, NK cell cytotoxic capacity, and immune escape in human and mouse systems, respectively. CLP, common lymphoid progenitor; CILP, common innate lymphoid progenitor; NKP, NK-cell precursor; iNK, immature NK; NK, natural killer cell.

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
