# Peer review of "The Role of miRNAs in Immune Cell Development, Immune Cell Activation, and Tumor Immunity: With a Focus on Macrophages and Natural Killer Cells"

_cells, 2019, doi:10.3390/cells8101140_

Round 1
Reviewer 1 Report
In this manuscript the Authors summarize the role of miRNAs in tumor immunity, a topic very interesting. The manuscript is well cut, however, it shows some issues and needs a carreful revision.
My concerns
There are data about the molecular mechanism involved in the up or down regulation of the miRNAs functioning in NK or macrofages development and tumor immunity?
What the difference between Figures 1-2 and Supplementary Figures?
Please check the References and Acknowledgments Sections.
Author Response
Point-by-point responses to reviewer’s Comments
Reviewer #1:
There are data about the molecular mechanism involved in the up or down regulation of the miRNAs functioning in NK or macrophages development and tumor immunity?
R: We thank for the valuable comments from the reviewer #1. To follow the question, the molecular mechanism underlined can be divided into two parts. One mechanism is transcription factor activation. As an example, PU.1 induces four miRs (mmu-miR-146a, mmu-miR-342, mmu-miR-338 and mmu-miR-155) expression to facilitate HSC differentiation into macrophages. Similarly, CCAAT/enhancer-binding protein alpha (C/EBPα) plays an essential role by direct binding to the promoter of hsa-miR-223 and induces hsa-miR-223 expression to promote granulocytic differentiation. Another mechanism is via cytokines activation. For example, TGF-β, a key mediator in TME, post-transcriptionally increases mature hsa-miRNA-1245 expression. This miRNA suppresses NKG2D expression, thereby blocks NKG2D-mediated immune responses in NK cells and supports TME. TGF-β also induces hsa-miR-183, to abrogate tumor cell-killing function of NK cell by targeting DNAX activating protein 12kDa (DAP12). These points and mechanism underlie are added into the manuscript (line 81-92 and line 221-224) .
What the difference between Figures 1-2 and Supplementary Figures?
R: We are sorry for the misleading. We don’t have any supplementary figures. There are two main figures and it is uploaded with text as a single file and separately added into the Figure section. Please only keep Figure 1 and 2, Table 1 and 2.
Please check the References and Acknowledgments Sections.
R: We thank for the helpful comment. We found error in references, so we updated the MDPI Endnote output style to correct the references. We also revised the author contribution and acknowledgments sections.
Reviewer 2 Report
The current manuscript focuses on two immune cell components in the tumor microenvironment (TME), namely macrophages and natural killer cells and specifically, on how miRNAs play a role on their development, migrations, differentiation, activation, and function as impacted by their localization and interactions with other components of the TME. The authors describe concisely and enumerate the miRNAs that have been identified in recent literature to play a role in these various processes. Two tables are included that list these miRNAs and their respective references, thus providing a useful resource for those interested in these molecules. Further, two figures are included which delineate the process where these miRNAs are implicated in both human and mouse cells, further providing a comparison of these processes in both species. Contrasting roles of some miRNAs as shown in literature were also discussed.
Comments:
1) It would be very useful to include in the conclusions section a discussion where these miRNAs might be useful particularly in therapeutic applications against cancer or inflammation that underlie their impact on the TME. Current investigations in literature that are geared toward these applications and their outcomes, if any, should be included to provide a rationale and significance to the data that has been enumerated throughout the manuscript.
2) There are numerous grammatical errors throughout the manuscript that makes it confusing at some points, and there are instances where references are missing or warranted. It is VERY IMPORTANT that the authors read the manuscript thoroughly to correct these errors or have the manuscript read and edited by someone with a good command of the English language semantics to correct these errors before the manuscript can be published.
Author Response
Point-by-point responses to reviewer’s Comments
Reviewer #2:
It would be very useful to include in the conclusions section a discussion where these miRNAs might be useful particularly in therapeutic applications against cancer or inflammation that underlie their impact on the TME. Current investigations in literature that are geared toward these applications and their outcomes, if any, should be included to provide a rationale and significance to the data that has been enumerated throughout the manuscript.
R: We appreciate for the valuable comments from the reviewer #2. Following reviewer’s opinion, we search the applications of miRNAs on clinical treatment. Briefly, the application can be divided into several parts, such as promoting anti-tumor apoptosis, suppressing cell proliferation or angiogenesis and activating immune response. For example, miR-21 and miR-29a, released from tumor-derived exosome, can directly bind to TLR7 receptor expressed on macrophages in the TME and then triggers inflammatory response to intensify tumor growth[1]. Therefore, blocking these secreted miRNAs is a potential application. Accordingly, we discussed the perspective role of miRNAs on therapeutic applications in the last paragraph. In our data, we picked out multi-functional miRNAs such as miR-181a, miR-150 and miR-155. These miRNAs not only regulate immune cell development and activation, but also participate immunity activation, thereby will be considered as good candidates on cancer treatment. In addition, we mentioned the one miRNA treatment may have risk or limitation as the same miRNA has different roles in different cell or cancer type. Our opinion is that using these candidate miRNAs as a cocktail may effectively improve the therapeutic effect.
There are numerous grammatical errors throughout the manuscript that makes it confusing at some points, and there are instances where references are missing or warranted. It is VERY IMPORTANT that the authors read the manuscript thoroughly to correct these errors or have the manuscript read and edited by someone with a good command of the English language semantics to correct these errors before the manuscript can be published.
R: We thank for the helpful point. Following the comment, we sent our manuscript to professional English editing company and fully revised our manuscript. Please check the attached certificate for the English editing service.

Reviewer 3 Report
The Authors undertake the important subject concerning the influence of miRNA on the differentiation and activation of selected components of the immune response (macrophages and NK cells). The main purpose of the review is to discuss the regulatory role of miRNAs on NK and Mf cells found in the tumor microenvironment.
Major concerns:
- Although one of the main purposes of the work is to describe the role of miRNAs on the activity of NK and macrophages in tumor microenvironment, the Authors addressed the problem very superficially. Described data is limited to the effect of virus infection-induced miRNAs on regulation of NK and Mf cell activity in TME. But even in this case information are disordered, incomplete and accidental. There is no information concerning the role of tumor-derived miRNAs on the immune cells, or regulation of the cells by miRNA derived from tumors which development is not depend on virus infection
- the article is a collection of basic information. The Authors do not attempt to discuss the results they described
- there is no section in which the perspectives of using the knowledge acquired so far would be discussed
- there is no convincing justification why NK and Mf cells were selected for this review.
- the information about real changes ongoing in NK and Mf cell functionality under influence of tumor-derived miRNAs would be valuable
- The Authors presents a lot of valuable data in tables, however they do not discuss them anywhere.
Taking all these points into consideration the article need to be strongly revised and completed before publication.
Author Response
Manuscript ID: cells-579730
Title: The role of miRNAs in Immune Cell Development, Activation and Tumor Immunity: Focused on Macrophages and Natural Killer Cells
Point-by-point responses to reviewer’s Comments
Reviewer #3:
Although one of the main purposes of the work is to describe the role of miRNAs on the activity of NK and macrophages in tumor microenvironment, the Authors addressed the problem very superficially. Described data is limited to the effect of virus infection-induced miRNAs on regulation of NK and Mf cell activity in TME. But even in this case information is disordered, incomplete and accidental. The information about real changes ongoing in NK and macrophage cell functionality under influence of tumor-derived miRNAs would be valuable.R: We really appreciate critical comments from the reviewer #3. We regret the description of our data is not very clear. Because there are nearly 200 miRNAs involved on the regulation of NK and macrophage activation or tumor immunity, it was difficult to explain the role of each miRNA clearly or make a simple conclusion. To overcome this, we divided all the related miRNAs into three parts and briefly presented each group of miRNAs.
Firstly, we show miRNAs working in macrophage polarization by cytokine stimulation (line 109-116). We not only indicated the top 4 miRNAs in mouse M1 and M2, but also showed 3 early-response miRNAs in mouse macrophages polarization.
Secondly, we added a new paragraph to prove tumor-derived miRNAs have important roles on the activity of NK and macrophages in TME (line 129-134). As an example, we mentioned miR-142 that suppresses macrophage differentiation, facilitates the immunosuppressive function in tumor. In contrast, we introduced miR-155 exerting opposite role by increasing the number of pro-inflammatory TAM, that in turn actively mediates antitumor immunity.
Thirdly, given that pathogen infection is an important factor for inflammation and tumor initiation, we described viral encoded, or viral infection-induced miRNAs in macrophages (line 135-142). Lastly, we explain the mechanisms by which miRNAs suppress tumor immunity, by blocking key factor expression in innate immunity activation (line 143-149).
Considering the length of manuscript, please understand that we only show two or three miRNAs in each part.
The article is a collection of basic information. The Authors do not attempt to discuss the results they described. There is no section in which the perspectives of using the knowledge acquired so far would be discussed. The Authors presents a lot of valuable data in tables, however they do not discuss them anywhere.
R: We appreciate for the critical point. As mentioned by the reviewer, our paper is a collection of all major miRNAs involved in macrophages and NK cells functioning, and we think that is one of the advantages of the article. Given that the number of miRNAs is too much and the complexity of miRNAs in each cell type, so we choose some miRNAs for discussion. To make a more reasonable discussion and provide our perspective clear, we now expand data discussion and divide it into three parts. These are miRNAs in macrophages (Figure 1 and Table 1, line 239-257), miRNAs in NK cells (Figure 2 and Table 2, line258-273) and comparing miRNAs working in both two cells(line 274-292). Besides, we also mentioned the risk and challenges for research uncovering miRNAs in immune cell function and developing miRNAs treatment.
There is no convincing justification why NK and Mf cells were selected for this review.
R: Following the reviewer’s comment, we had explained the reason why we choose NK and macrophages in the introduction. We appreciate for the valuable comment to make this part clear.
Round 2
Reviewer 3 Report
The Authors have completed the manuscript with some information and re-written some its sections. Language style and grammar were significantly improved. All the changes made the manuscript more readable and coherent.
It is not clear why in the last column of Table 1 is the part concerning general immunity activation. What about the list of miRNA related to tumor immunity? It should be completed, if not explained in the text.
Author Response
We appreciate to the Reviewer #3 for additional helpful comment. We agree that the title of last column in the Table 1 did not fully describe its contents. Following the reviewers' suggestion, we changed it to "Relation to tumor immunity".
Thank you again for the comment.